# Digitalization in Medicine: Are German Medical Students Well Prepared for the Future?

**DOI:** 10.3390/ijerph19148308

**Published:** 2022-07-07

**Authors:** Heiko Sorg, Jan P. Ehlers, Christian G. G. Sorg

**Affiliations:** 1Didactics and Education Research in the Health Sector, Faculty of Health, University of Witten/Herdecke, 58455 Witten, Germany; jan.ehlers@uni-wh.de; 2Department of Plastic and Reconstructive Surgery, Marien Hospital Witten, 58452 Witten, Germany; 3Department of Management and Entrepreneurship, Faculty of Management, Economics and Society, University of Witten/Herdecke, 58455 Witten, Germany; christian.sorg@uni-wh.de

**Keywords:** medical liability, medical school, patient contact, profession, web-based learning

## Abstract

The German healthcare system is facing a major transformation towards digitalized medicine. The aim was to find out the attitude and the degree of preparation of upcoming medical professionals for digital medicine. By means of an online survey, medical students from 38 German faculties were asked about different topics concerning digitalization. Most students (70.0%) indicated that they had not had any university courses on digital topics. Thus, only 22.2% feel prepared for the technical reality of digitalized medicine. Most fear losing patient contact because of digitalized medicine and assume that the medical profession will not be endangered by digitalization. Security systems, data protection, infrastructure and inadequate training are cited as the top problems of digitalization in medicine. Medical students have major concerns about incorrect decisions and the consecutive medicolegal aspects of using digital support as part their treatment plans. Digitalization in medicine is progressing faster than it can currently be implemented in the practical work. The generations involved have different understandings of technology, and there is a lack of curricular training in medical schools. There must be a significant improvement in training in digital medical skills so that the current and future healthcare professionals are better prepared for digitalized medicine.

## 1. Introduction

With demographic change, the German health care system is faced with the task of providing medical care of at least the same quality in the face of increasingly multimorbid patients on the one hand and a growing scarcity of resources in the form of a shrinking number of contributors on the other [1]. This is aggravated by a significant shortage of physicians and specialists in many medical fields, which could challenge the provision of comprehensive care in the future, especially in rural regions. At the same time, the demands of patients are increasing, who currently generally obtain information in advance of their diagnosis on the internet [2,3,4], and in many cases expect participatory decision-making in the treatment process.

Digitalization, which has already made its way almost unstoppably into many areas of life, is often seen as at least part of the solution to future problems by politicians and health insurers [5]. Digital applications such as the HIS system, radiological teleconsultations and health apps are already part of everyday life for many physicians [6]. Technical progress and the digitalization of communication, workflows, and processes, as well as an ever-increasing volume of data and its networking, offer new opportunities for diagnostics, therapy, and prevention. However, numerous unanswered questions, for example in the area of data protection or technical infrastructure, make it clear that digitalization does not appear to be a foregone conclusion, as illustrated, for example, by the difficult debate over electronic health cards in Germany over the last few years [7,8]. These hurdles also address the question of the extent to which patients can be transferred to digital medicine at all in the current time and situation. Age, gender, place of residence, education, ethnicity, race, and the financial status of patients must be considered as well [9,10,11,12]. The physicians’ view of digitalization in medicine as a user and multiplicator will play a further decisive role in the implementation of any such innovations. Major concerns of healthcare professionals apply not only to the security of systems and infrastructural requirements [13], but also in particular to the preparation for the coming digitalized world [14], the changes in the workplace, the medical practice and the transformation of the medical profession itself [15].

Although most of today’s physicians and especially the next generation are much more familiar with computer-based technology than their predecessor generations [16,17,18,19,20], the implementation of digital medicine and its digital teaching and learning formats in everyday life requires appropriate training and continuing education programs for healthcare professionals [21,22]. At the same time, it must not be disregarded that digital teaching or digital learning is an adequate means of increasing knowledge for everyone [18]. Thus, the intensive private use of digital media by medical students cannot necessarily be inferred as a meaningful application during studies [23]. Furthermore, even lecturers with little or no didactic training and varying degrees of rhetorical ability will not automatically be excellent e-teachers [24,25]. During the pandemic, the teaching conditions at universities did not change significantly. There is still a difficult balance between research, teaching and patient care for lecturers and increasing student numbers, and there are less available patients for teaching. Consequently, digital teaching formats and e-learning can, indeed, be part of the solution.

In this context, the transfer of knowledge in medicine combined with the question of the boundaries of a purely digital medicine should also be part of the focus. At this point, it is necessary not only to talk about the mere establishment of as many programs as possible, but also to think about meaningful definitions and configurations of digital teaching [23,26,27,28]. Medical students, in any case, are and have been positively disposed towards this development, especially due to the fact of gaining more independence and higher personal responsibility in their studies. An intelligent, balanced, and complementary mixture of reduced classical teaching [29] and self-study using modern e-learning technologies with curricular integration could help to address this problem [24,30,31].

Despite the many issues that remain unresolved, there is still hope that the digital transformation of medicine, through technological capabilities, will improve clinical outcomes to enhance people’s quality of life and democratize healthcare [32]. However, with a science that is based on a physical examination and treatment of the patient, the reasonable boundaries of digital learning must also be discussed. Whereas digital learning formats were still carried out rather sparsely and awkwardly before the year 2020, the situation today is quite different. In no time at all, entire semesters have been converted to digital formats and all participants have been “forced” into digital teaching. Many national and international groups of authors have since described whether and how well this has been achieved [33,34,35,36,37].

Based on the pre-pandemic situation, the question arose as to how well-prepared medical students are for digitalization and digital medicine per se and how they generally feel about the changes, benefits, and disadvantages of it. This study will therefore be dedicated to this topic and will provide appropriate clarification.

## 2. Materials and Methods

This study was conducted as a descriptive cross-sectional study that was part of study arm XI of the Careers in Medicine (KARiMED) study (www.karrierestudie.de, accessed on 1 June 2022). In preparation for the study and the questionnaire required for it, the three authors of this study and two other members of the KARiMED team joined forces in the form of an expert panel. Within the framework of this expert panel, the main problem areas of digitalization in medicine implied by the authors were first presented. Within these issues, urgent questions were then elaborated on, discarded, compiled, and finally prepared for the online questionnaire. A general part supplemented the questionnaire at the beginning to better describe the cohort and to be able to perform group analyses. Dichotomous questions (yes/no), Likert-scale questions (five answers), and open-ended questions were used for this purpose. The evaluation of the free-text answers was assigned to the given answer categories in accordance with the content analysis according to Mayring [38], as far as this was justifiable in terms of content. The decision on this was made by two independent assessments from the team of authors. Otherwise, the remaining answers were evaluated as an independent category: “other”. Furthermore, there were dependent questions that only served to further classify the corresponding answer in the case of a specific answer (yes/no). Thus, not all questions were shown to all participants. This resulted in the following five question subsets:▪a general section with questions about the person (age, gender, place of study, period of study, affinity for digital media, use of specific hardware/software): 15 questions▪digitalization at the university: 5 questions▪digitalization of medicine and future: 47 questions▪medical activity and professional status in digital medicine: 7 questions▪ethics and law in digital medicine: 12 questions

The online questionnaire thus contained a total of 86 questions.

*Survey*. The survey for the study presented here was conducted as an online survey using LimeSurvey software on a dedicated server in the period from 1 March 2018 to 31 May 2020. An anonymized questionnaire was developed to collect the data. The general part contains socio-demographic and study biographical characteristics of the students, with questions about gender, age, progress of studies as well as place of study. The special part is made up of the above-mentioned topics.

The study was conducted in accordance with the Declaration of Helsinki and approved by the Institutional Review Board of the Hannover Medical School, Hannover, Germany (No. 1862-2013).

*Recruitment of participants and inclusion criteria.* Study participants were recruited from all 38 medical faculties and universities in Germany. The survey was announced via social networks, the student councils and student council mailing lists of the medical faculties, the Medical Education Working Group of the Federal Association of Medical Students in Germany (bvmd), and by e-mail. Inclusion criteria in the study were the current study of human medicine and the study location being in Germany. Students of human medicine in their first to 12th semesters were surveyed in order to obtain a broad opinion from the preclinical and clinical study stages. Excluded from the survey were students of other study programs, including dentistry and veterinary medicine, students of human medicine from other countries, and those participants who answered the questionnaire incompletely.

Quantitative analysis was performed by descriptive statistics using the program R (version 4.1.0; 18 May 2021).

## 3. Results

*Participant collective and digitalization at the university.* A total of 434 medical students (66.4% female) participated in the survey. There were no exclusions regarding the wrong study subject or foreign students. A total of 255 questionnaires were excluded from the analysis due to incomplete responses. Students from a total of 27 universities in Germany took part in the survey. Further description of the study cohort is shown in Table 1. Most survey participants reported not receiving any teaching units on digitalization at their own faculty (69.1%; *n* = 300). Only 10.4% (*n* = 45) answered “yes” to this question, with 55.6% of these being optional and 66.7% being mandatory courses (multiple answers possible). However, most medical students (51.1% (*n* = 23), who answered “yes” on receiving any teaching units on digitalization) that these courses did not prepare them for the technical reality of working as a physician in a few years. Only 22.2% felt that they were adequately prepared. However, at 64.0%, the survey participants clearly want courses on digitalization as part of their studies. The majority of medical students owned a smartphone (99.5%) and a laptop/notebook (95.6%). An app from the health sector (e.g., for sports tracking, gymnastics, training) had 70.9% on smartphone or tablet computer, which is used “occasionally” (35.7%), “often” (17.2%) and “regularly” (20.1%). The medical students who are currently predominantly members of Generation Y and Z are well acquainted with technology (25.4% “agree”; 32.5% “tend to agree”; 27.9% “partly/partly”) and working with a computer does not cause them any problems (problems working with a computer: 1.4% “agree”; 4.6% “tend to agree”; 11.8% “partly/partly”). While the survey participants have a clear idea of what is meant by the term telemedicine (known by 64.5%), this is not the case for the terms “e-health” (known by 31.3%) and “Big Data” (46.1% of respondents).

*Description of the clinical activity in digital medicine.* When asked whether participants had thought about working in a more digitized medicine before this survey than is the case today, 75.9% answered “yes” (*n* = 410; 43 participants could not assess this question). Interestingly, however, most medical students do not want to work primarily digitally (e.g., telemedically) later. Thus, 318 participants (73.3%) stated that they would only do so “reluctantly” or “very reluctantly,” 93 “partly/partly” (21.4%), and only 23 “gladly” to “very gladly” (5.3%). We also asked whether diagnostic tools could take over medical activities in the outlook. 34.3% of the medical students “tend not to believe this” and 7.8% even “definitely not”, while 29.0% tended to agree and 6.5% even said “definitely” (the remaining 22.4% said “maybe”). In this context, most survey participants fear losing contact with patients because of digitalization (66.6%). The necessary prerequisites from the perspective of the study participants for physicians with regard to digitalization in medicine are shown in Figure 1.

*Future of the medical profession in digital medicine.* Digitalization will change the medical profession in the future. When asked whether the medical profession is at risk from digitalization in the future, 3.0% answered “yes”, 8.0% answered “rather yes”, 10.4% answered “maybe”, 35.3% answered “rather no”, and 43.3% answered “no, definitely not.” Most respondents also do not believe that the profession of physician will be lost because of digitalization, insofar as more functions would be taken over by non-physician professions as a result (66.1%). The affinity of the survey participants for technology is also reflected in the question of whether there are concerns about not being able to cope with the technical programs and requirements. This is “rather not” or “not at all” the case for 72.4% of respondents. Like the response to the prerequisites, the top problems of digitalization in medicine are cited as backup systems (e.g., in the event of system failure; 67.1%), data protection (66.4%), training (59.4%) and infrastructure (59.4%) (Figure 2).

*Ethics and law in digitalization.* When it came to the issue of ethics and law in the digitization of medicine, medical students were rather concerned about the security of the system and the use of data. Thus, 87.5% saw it as “rather problematic” or even “very problematic” to be able to protect themselves against external hacker attacks. At 85.5%, monitoring by health insurance companies and 85.4% data protection was also seen as major problematic issues in digitalization of medicine by medical students (Figure 3). Whether data protection can be regulated in the near future in such a way that the use of patient data confers an advantage for treatment showed a rather heterogeneous picture (“yes in any case” 12.2%; “probably yes” 34.6%; “perhaps” 29.0%; “rather not” 21.0%; “no, in no case” 3.2%). In contrast, 90.3% of respondents stated that digitalization is also based on commercial aspects. Other critical issues related to digitization in medicine are summarized in Figure 3.

In the context of diagnostics, there could be improvements using Big Data, for example. In the future, the question of responsibility for possible misdiagnoses and the resulting incorrect treatment will arise both ethically and in terms of medical law. In this context, we asked the survey participants three questions on the topic of law in digitalized medicine. When asked how to proceed if their own diagnosis diverged from the computer diagnosis, 85.0% answered that they would seek the advice of a colleague (9.4% trusted their own diagnosis; 0.9% trusted the computer’s diagnosis; 4.6% could not assess this). When asked about liability in the event of incorrect diagnosis of a nevus by a digital skin scanner and subsequent death years later, 56.5% responded that the physician would still be liable (32.9% could not assess this; 6.5% held the manufacturer of the tool liable; 2.8% held the clinic liable; 1.4% held the supervisor liable). Even in the case of a significantly faster death due to an incorrect assessment of a diagnostic tool for the diagnosis of myocardial infarction and the death of the patient after hours on discharge, because the tool did not see a constellation of myocardial infarction, as many as 64.1% answered that the liability lay with the attending physician (29.0% could not assess this; 4.6% believed that the manufacturer is liable; 1.8% believed that the hospital is liable; 0.5% believed that the supervisor is liable). The survey participants were also asked if they were afraid of incorrect decisions as a result of computers making different diagnoses. Thus, 26.5% stated that this was “true”, 33.2% that it was “somewhat true” and 19.6% that it was “partly true”. Only 15.4% said that it “rather does not apply” and 5.3% stated that it “does not apply”. This aggravates even further with the fear of possible regress or legal proceedings because the physician has made a different decision than a computer program. Thus, it is “true” for 28.6%, “rather true” for 38.5%, “partly true” for 18.0%, and “rather not true” for only 9.9% and “not true” for 5.1%.

## 4. Discussion

This national cross-sectional survey of medical students on the topic of digitalization in medicine at medical faculties in Germany is intended to provide an overview of the pre-pandemic situation and evaluate the participants’ views. The digitalization of medicine and medical teaching has undoubtedly significantly increased because of the pandemic. Digital teaching formats are now available to 97% of German-speaking medical students [34]. However, this development came very spontaneously, and medical students felt that universities had responded only moderately well to this situation [34]. While there is now a great deal of research on the topic of digitalization in teaching, results on the general attitudes and concerns that digitalization brings to medicine per se were not available. This is where this study with data directly before the pandemic situation can provide information and debate.

As a limitation of this study, some points need to be discussed at this point. Despite multiple invitations and reminders as well as an optimized questionnaire design, 434 questionnaires were included in the study. The methodology for creating and using questionnaires as instruments of scientific surveys are well established in science, and online surveys have become an indispensable part of modern communication and science [39,40,41]. An online survey can be implemented quickly. This applies not only to the invitation of potential participants, but also to the evaluation of the data, which is available immediately after processing the questionnaire. Furthermore, the online survey is cost-effective because no external institute needs to be involved. Due to the anonymity, an online survey also usually provides honest answers and thus increases the quality of the data. Despite the supposedly low participation in the survey, it can therefore still be assumed that the data have a corresponding content in the sense of the assumptions for a large total of those affected, since responses were received from a total of 27 of the 38 German medical faculties or universities (71.1%).

A further limitation of the study is clearly the timing of the investigation, which, with the end of the inclusion of participants, heralded the beginning of the pandemic-related digital transformation of teaching at national and international universities. Thus, the picture presented here clearly reflects the situation prior to the COVID-19 pandemic, where digital teaching still was a rare case, as also indicated in the respondents’ answers. A comparative survey at the present time, or at the end of the pandemic, would certainly be of interest. In this context, the COVID-19 pandemic has brought about changes in many areas with the reduction in contacts as well as the lockdowns carried out. Particularly in the area of teaching, lockdowns created new digital formats and online platforms which, for example, made it possible to study medicine almost fully in the “home office” [36]. The faculties have managed to redesign their curricula in such a way that a large part of the teaching and exams could be held digitally [42]. Furthermore, video-based formats, online chats and patient-physician telehealth applications have gained more importance, satisfying teaching staff to better meet medical students’ needs [30,43]. However, although faculty seemed more satisfied about the outcome of their teaching, they were generally more stressed compared to their personal and professional lives as clinicians, presumably because of the re-design of their teaching templates to digital formats [43].

Similarities were found in other countries as well [33,44,45,46,47,48,49]. The online teaching format ensured continuity in study progress despite the pandemic and was therefore predominantly considered to be effective. Although it was more frequently described that face-to-face education would be better, the students were able to get used to digital teaching and, subsequently, after appropriate training, even developed a time saving and performance improvement. Interestingly, students who had already used online formats generally coped better with this situation. However, training and improvement in the handling of the new formats was necessary on both sides, as there were probably only inadequate, partially fragmented solutions at the beginning, this being attributable to the pace of development. As the lockdown progressed, lecturers and students were able to adapt to each other and the faculties provided the appropriate technical and infrastructural resources. However, across countries, medical students felt that the practical and individual learning needs of students should be considered and attempted to be addressed in the implementation of online medical education [33,44,45,46,47,48,49].

Web-based learning obviously has both advantages and disadvantages [28,50,51]. Individual learning independent of time and place, flexible scheduling, overcoming distance and time barriers, economies of scale, and novel teaching methods can be cited as clearly advantageous. The lack of student interaction with faculty, social isolation, time to prepare and the need to remain current with regard to digital teaching formats, upfront costs for infrastructure, and technical problems can be considered as drawbacks [50,51,52,53].The range of courses available at universities for this purpose is broad, diverse and therefore difficult to oversee [23,50,51,54,55,56,57,58,59,60,61,62,63]. Moreover, students’ place of study, origin, previous exposure to digital teaching content, and gender need to be considered [49,56,64,65]. Each university, faculty, and even discipline employs different forms of digital teaching. For example, more virtual reality applications are used in surgical and intensive care specialties [55,60,66]. Interestingly, in surgery, the classical main lecture was still considered quite important before the pandemic [29]. Exactly this has now changed, as the most frequent transformations in digital teaching take place in the context of obligatory lectures in the individual disciplines [61,62]. Bedsides teaching, thus, is still difficult to adapt to a digital format [62]. In contrast, communication skills with virtual patients can be well performed web-based and successfully trained [67]. Apart from purely clinical training, acquisition of a scientific competence for German medical students has also recently been on the curriculum of the National Competence-Based Learning Objectives Catalogue. Although digital teaching in otolaryngology was not exclusively evaluated positively by the students in the study by Rauch et al., their subjective scientific competence did improve, which was used in the context of an online scientific seminar on writing scientific articles [59,68].

In summary, the information from this study and the current knowledge is crucial for future work in these areas to support and address various gaps in the field and for comparing studies (pre- to post-pandemic). This brings to the forefront the opinions and concerns of future medical professionals who will eventually use appropriately verified technological developments in their subjects. Understanding the perceptions of current medical students can shape the course content and future directions for developing necessary training and processes for medical students that can equip them well to manage the increasing digitalization of medicine. Therefore, a sensible mixture of digital teaching with face-to-face units in the sense of blended learning would be a reasonable and feasible strategy to implement the digital transformation of the medical curriculum in Germany in the long term and makes students better prepared for further pandemic situations or similar circumstances [24,50,51,54,57]. As David Cook has stated, “*Like blackboards and slides, WBL [web-based learning] is a powerful tool—but only a tool—that if used wisely can greatly facilitate learning*” [50].

Two further aspects also have to be discussed when it comes to digitalization in medicine. On the one hand, digitalization can improve diagnostics and therapy, in particular by increasing the efficiency of medical activities in principle [69]. On the other hand, this increase carries the potential danger of the substitution of medical services and, consecutively, even the loss of the entire profession [70]. In this debate, it seems unquestionable that the role of physicians will change in the future due to digitalization in medicine [71] and will probably be composed of a combination of the above. For example, the sensible use of smart algorithms and Big Data offers doctors the opportunity to simplify and accelerate medical diagnosis and therapy planning. This neither restricts physicians’ freedom of decision, nor does it lead to a loss of medical services due to digital technology. On the contrary, in the future physicians will act as interface managers between their own experience, technological, algorithm-based data processing, support systems and patients [72,73,74]. Since knowledge within medicine is expanding so dynamically in the meantime, it will also become a medico-legal problem for physicians in the future to not be supported by digital systems. Digital support can, for example, prevent errors in the event of inadequate information processing and thus prevent harm to the patient [23,75].

In the study presented here, the fear of losing patient contact through digitalized medicine was frequently expressed by the students in this context. For decades, however, physicians across all disciplines have been complaining about the increasing administrative workload and paperwork in the course of their work, which significantly limits the actual medical activity. In this regard, digitalization could create rationalization opportunities through secure communication and documentation, value-added potential by offering new medical care options, and potential through the emergence of artificial intelligence, which changes the nature of medical work itself [76]. The use of digital infrastructures and artificial intelligence, therefore, opens many possibilities in terms of saving time by absorbing the burdensome administrative and organizational activities and to allow the doctors more time for patient contact. It is therefore up to the physicians themselves to use digital technology for their own benefit so that depersonalization does not occur [77]. Here, many digital technologies and software can already be used today to facilitate communication, scheduling, or file management. Even though the medical profession is generally open to the use of digital solutions and technologies, the reluctance to develop a digital infrastructure in German hospitals so far is primarily due to the strict data protection requirements and guidelines relating to IT security in Germany [78]. Certainly, the odd medical or non-medical job may be affected by digital transformation. It can also be assumed that diagnostic disciplines, such as radiology, microbiology, and pathology, will probably move closer together. Physicians will continue to be needed in the future, but their role will change. While digital systems will provide support in diagnostics, the physician himself will be in demand for the interpretation and communication of results. However, this also holds the opportunity for the evolution of new occupational fields within medicine, e.g., as specialists in digital medicine, specialists in hospital communication, healthcare content managers, or medical-technical assistants in the field of Big Data and e-health [79,80]. In his editorial in 2018, Ertl already called for a medical informatics specialist as a communicator between medicine and informatics [81]. He also noted that the acceptance of new digital technologies by physicians can only be increased if they are actively integrated into the development process [81]. While a specialist veterinarian for information technology position was established in veterinary medicine in Germany some years ago [82], this development has now been followed in the USA with the creation of a position specializing in clinical informatics [83].

Medical students in general, as reflected in this study, currently have no job-related concerns about digitalization [84]. Medical students, however, have very controversial attitudes towards the medical field of radiology regarding possible changes due to digitalization [85,86,87,88,89]. As members of Generation Y (birth cohorts 1980–1995) and Z (birth cohorts 1995–2010), they are highly open to new digital technologies and media and are very well versed in this area [90,91]. It is interesting that there seem to be different attitudes and perceptions in different countries. While Australian students are positive about the use of mobile device applications in the medical curriculum [92], Chinese students tend to be less active users of mobile devices for learning activities [93]. Although digital content is already offered at some universities, almost 70.0% of medical students stated that they had no teaching units on this subject prior to the changes that resulted from the COVID-19 pandemic. As a result, students do not feel that these courses prepare them adequately for the digital aspects of their future work. In 2006, students were found to be still very differentiating about their skills in medical informatics in this regard. In aspects they have grown up with (i.e., using online resources, software to make graphs, and prepare presentations), their abilities have been rated highly. Lower abilities have been assessed with regard to using clinical information systems, accessing databases of clinical information, and knowledge of advocacy resources [94]. Consequently, medical schools should teach digital skills that may be necessary for the future work of physicians. In a Delphi-process, a panel of experts identified a total of 40 recommendations that faculties can use to design their digital health curricula [95]. For example, they recommended comprehensive introductions to topics that were also part of the survey presented here, such as health data infrastructure, clinical decision support systems, and digital ethics [95]. The wide range of the field also contained the recommendation to teach basic concepts of artificial intelligence (AI) for health issues [95]. AI presents a unique situation here, as there are qualitative differences for issues of explainability, health equity, and data security [96,97,98,99]. While students are to some extent well aware of the possibilities and limitations of AI [85,100], it has also been shown that the absence of such content in the curriculum leads to helplessness, anxiety, and altered career choices in dealing with this topic and should therefore be urgently included as a digital literacy in the compulsory medical school curriculum [84,88].

The importance of digitalization in healthcare for university education has in the meantime been recognized at medical faculties. Yet there is also agreement that students are not sufficiently prepared for digital patient contact and lecturers do not have sufficient competence in teaching digital learning content [101]. Thus, in a rapidly developing technology, the curricula of German universities have apparently not yet been adapted to the digital transformation. The description of digital medical competencies is still missing in the Master Plan Human Medicine 2020 [81,102], although necessary digital prerequisites for the medical profession have long been proposed by medical informatics societies [21,103]. As a first initiative, the 2021 version of the National Competency-Based Learning Objectives Catalog for Medicine (NKLM 2.0) in Germany now includes the critical and patient-centered use of information technologies through digital diagnostic and therapeutic support systems, mobile health/smart devices, and Big Data in its catalog of medical science skills [104]. The lack of preparation or further training in digital topics is not evident among students alone [94], but plays a major role among those already actively involved in medicine. It has been shown that physicians have considerable concerns and deficits, particularly with regard to the possibilities, risks and limitations of digitalization [83]. The high level of skepticism toward the use of new technologies was demonstrated, for example, in the use of mobile health apps as a possible prescription therapy in Germany. Physicians indicated substantial concerns (i.e., insufficient information, reimbursement, medical evidence, legal and, technological uncertainties) about the prescription and use by the patient [14]. In this case, uncertainty or even refusal to engage with digital tasks not only squanders the potential of these technologies and improvements in patient care but can also be accompanied by adverse health effects for medical staff [105,106,107]. Against this backdrop in particular, training will be necessary for the successful implementation of the digital transformation. One challenge is that training must establish the basis for continuous professional development in dealing with data and technologies, which does not yet exist adequately today. The paradigm shift in training must therefore replace the previous simple positivism with a method-centered and science-oriented attitude and competence development. It will no longer be enough to learn digital technologies in medical school, as digitalization itself must become part of the education [23].

Furthermore, the field of digitalization in medicine is already so complex that physicians can lose track of it. This involves not only digital platforms for documenting patient data, telemedicine, and digital communication options, but also the teaching of media skills, the use of databases and information systems, and the application of digital decision support systems. Compared to other industries, these digital possibilities are still rarely used within the healthcare sector in Germany and are far from being applied across the board [108,109]. For example, the degree of collaboration (industry maturity) in the healthcare sector is still an estimated 10 to 15 years behind other industries such as retail or the automotive industry [110]. Another problem is that physicians hardly benefit from digital health. The digitalization of healthcare has so far been primarily aimed at improving things for patients and the healthcare system. However, the benefits and impact on physicians have scarcely been taken into account. Solutions are only just being developed in this area.

This change, on the other hand, is not taking place among physicians alone, but will inevitably change the physician-patient relationship. As already mentioned in the introduction, patients will want to play an active role in managing their therapy through a wide range of digital services and will therefore demand competent advice from their physicians and the ability to handle the new technologies with confidence [111]. In this context in particular, not only is the ongoing training and continuing education of the medical professionals necessary, but also the definitive integration of informatics and digital teaching and learning content in a medical degree program of tomorrow [112]

The digital fascination, however, still has its limits, at least in Germany [7]. In addition to the infrastructural requirements, data protection and security systems, there are other problem areas in digital medicine. On the one hand, there are questions about how to deal with the huge data sets obtained every day and their unused potential. On the other hand, there are ethical concerns in the sense of incorrect treatment of patients due to misdiagnosis, for example, using decision support systems. Here, however, the medical students are quite clear, as they predominantly stated that the responsibility of making a correct diagnosis ultimately still lies with the attending physician. But this is precisely what the medical students are also uncomfortable with, as not only are just under 60% of the participants worried about incorrect decisions being made by computers, but even more (67%) are afraid of the then threatening recourse and legal disputes. As a result, the digital transformation in medicine is also bringing new legal problems into focus, the solutions to which are already being tackled by the legal system. In particular, adjustments will have to be made to medical liability and doctors’ duties of care, and it can be assumed that patients will also be assigned greater joint responsibility [113].

## 5. Conclusions

The digital transformation of the healthcare system is already in full progress and will inevitably and fundamentally change the medical profession without replacing it. This poses enormous changes both for the organization in hospitals as well as universities and for the physician’s job profile in particular. However, it is apparent that developments are progressing faster than they can be implemented in practical medical work. The speed of development in digital medicine does not, of course, give rise to too many uncertainties among medical students regarding their future careers as physicians. For their clinical work, however, this seems to be different and not just a problem for students alone. This is not only due to the different understanding of technology among the generations involved in medicine, but also to the lack of curricular training in the current design of medical studies in Germany. The obstacles that digital medicine currently still brings with it can only be overcome by significantly improving education, training, and continuing practice in this area. This would better prepare current and future physicians for digital transformation by losing the fear of Big Data, using software systems for administration and communication safely and successfully, and facing potential digital failures with confidence.

## Figures and Tables

**Figure 1 ijerph-19-08308-f001:**
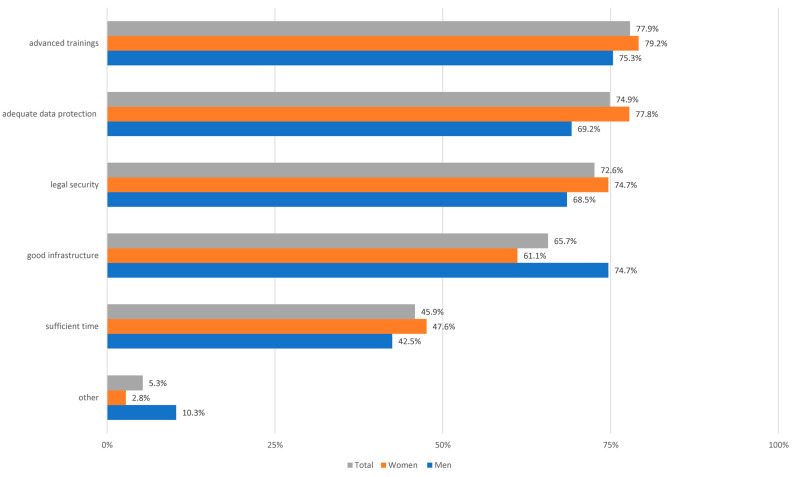
Description of the necessary prerequisites for physicians in the digitalization of medicine from the perspective of the study participants (multiple answers were possible; *n* = 434).

**Figure 2 ijerph-19-08308-f002:**
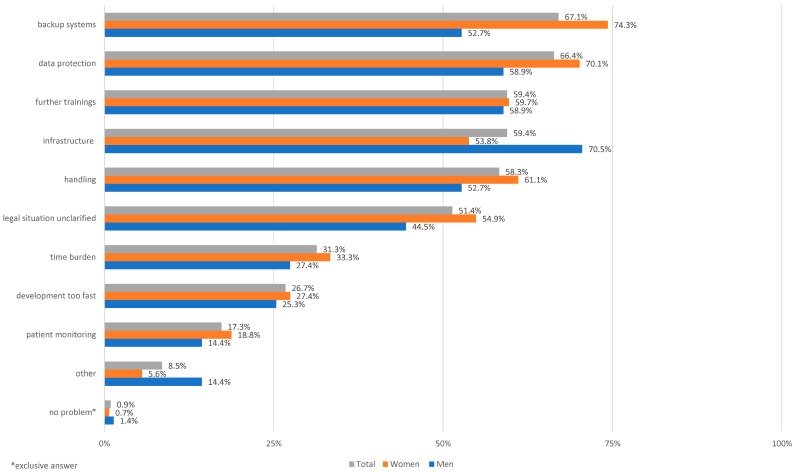
Illustration of the medical students’ assessment of potential problems of digitalization in medicine (multiple answers were possible; *n* = 434). Infrastructure, i.e., the necessary software and hardware to use respective tools or platforms in digital medicine; handling, i.e., confident handling of the digital applications; patient monitoring, i.e., data-based monitoring of patients (laboratory values, diagnostic results).

**Figure 3 ijerph-19-08308-f003:**
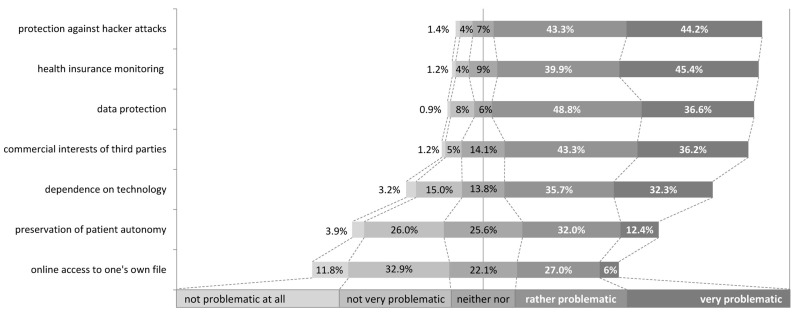
Presentation of the assessment of the medical students (*n* = 434) on the question “How problematic do you think the following points are with regard to digitalization in medicine?” on a 5-point Likert Scale.

**Table 1 ijerph-19-08308-t001:** Description of the KARiMED study participants from study arm XI (*n* = 434).

General Information, *n* (%)	Total	Female	Male
Participants, *n* (%)	434 (100.0%)	288 (66.4%)	146 (33.6%)
Age in years, mean (±standard deviation)	25.1 (±4.3)	24.6 (±3.8)	26.1 (±5.0)
Pre-clinical (1st–4th semesters), *n* (%)	97 (22.4%)	65 (22.6%)	32 (21.9%)
Clinic (5th–10th semester), *n* (%)	263 (60.6%)	177 (61.5%)	86 (58.9%)
Practical year (11th–12th semester), *n* (%)	74 (17.1%)	46 (16.0%)	28 (19.2%)

*n* = number.

## Data Availability

Data is archived and can be obtained by the KARiMED Study Team.

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
