# Peer review of "Digitalization in Medicine: Are German Medical Students Well Prepared for the Future?"

_ijerph, 2022, doi:10.3390/ijerph19148308_

Round 1
Reviewer 1 Report
Digitalization in medicine is certainly the future. In this paper, the authors present a study on the readiness of the German Medical students for the transformation to digital medicine. This is an interesting and potentially useful investigation. However, this study provides an inadequate explanation of the background which makes it difficult to value the importance of this work.
The reviewer strongly recommends a revision on the following facts:
1. The authors should clarify the significance of this study in the introduction in contrast with the existing research. The scope and the boundary of medicine digitalization should be defined in this section as well. This will help readers to understand the scope of the work and differentiate it from digital pathology, digital health, and others.
2. “How were the survey questions selected?” – should be explained in the methodology.
3. The findings of the study can be presented using cross-tabulating subgroups in the result section.
4. This study should share some insights into how the education materials and pieces of training can be designed to prepare the student for the transformation.
Reviewer 2 Report
1. This research lacks a summary of the existing literature and a review of the recent research in the last two years. More recent research papers can be added to the references.
2. The purpose of this paper is highly unclear, as it is clear from the introduction that this paper wants to explore the acceptance and general feelings of medical students towards digital healthcare. However, the discussion section discusses the two sides of digital healthcare, the problems that digital healthcare can cause and other findings that are irrelevant to the research topic.
3. As the authors use questionnaires in the analysis, one would appreciate the feedback on the structure of the questionnaire, the number of items on every scale, and the overall number of items in the questionnaire applied in the survey.
4. How do the authors measure medical students' attitudes and feelings towards the future of digital health technology? Please add the specifics of the question items.
5. The study lacks a theoretical contribution and seems to lack practical implications by simply exploring medical students' acceptance of the future of digital healthcare.
6. The degree of digital healthcare in Germany seems to be relatively well established, so does the subject of this paper lack a certain degree of innovation?
7.There are inconsistencies in the authors' conclusions. In the discussion section, the authors state that the study's findings show that medical students have no concerns about the future of digitalisation in their profession. However, in the conclusion section, the authors suggest that digital fears of future doctors could be dispelled by improving education on digital healthcare. So what are medical students' attitudes toward the future of digital healthcare?
Reviewer 3 Report
Congratulations on submitting the manuscript titled “Digitalization in medicine: Are German medical students well prepared for the future?”. The objective of the work presented in this manuscript was to understand the perspective and preparedness of medical students for digital medicine. This objective is well articulated in the abstract and introduction. The authors report findings from an online survey conducted among students at medical faculties and universities in Germany during the period of March 1, 2018, to May 31, 2020. The results shed light on the perception of students on topics of prevalence of digitization related coursework, how well these courses equipped them, acquaintance with technology, prerequisites for physicians in digitalization of medicine (DoM), impact of DoM on the medical profession, major challenges faced by DoM, and ethic and Law implications of DoM.
The manuscript is well articulated, and the information has a logical flow to it to move through different sections of the manuscript. Even though the survey was conducted only at the medical faculties and universities in Germany, their findings can also be applicable to and can provide input for similar assessments at number of other faculties and universities.
This information is very crucial for guiding some of the future work in these areas to support and address various gaps that exist in the field. It brings to forefront the opinion and concerns of future medical professionals who will eventually be utilizing appropriately vetted technological developments in the field of medicine. Understanding the perceptions of current medical students can shape the course content and future directions for developing necessary trainings and processes for medical students that can equip them well to navigate the increasing digitalization of medicine.
Comments:
Please find the comments for the manuscript in this section.
Line 30: Further clarity into the meaning of ‘more limited patients’ in the sentence will be helpful
Line 34: Further elaboration on what is meant by the ‘patient’s performance’ will be helpful.
Line 58: The objective was to assess the perception and the degree of preparedness of medical students which were evaluated through the survey designed to dig deeper into selected areas which were identified to be relevant. It would be helpful for readers to get insights into the sources and the rationale used for determination of questions and choices used in the survey. It will help readers to gain a better understanding of the process that was followed and can aid the development of similar studies in other regions.
Lines 62-64: It would be beneficial to add the number of questions included for each category. Additionally, providing information about whether the questions had fixed options or if some of them include free text input will give better clarity on the design of the survey questions.
Lines 65-66: It would be useful to provide information if this is an exhaustive list of all the medical faculties and universities in Germany that were surveyed originally or if it were the list after the application of exclusion criteria since one of the options was incomplete surveys.
Lines 99-106: It is an interesting finding that the participants were reluctant to work digitally and even fearful of losing contact with the patients. Was this reluctancy due to the fear of losing contact with patients or was it also due to the lack of necessary prerequisites? It is unclear whether the options surveyed as necessary prerequisites were used to identify the considerations that influenced the response to the previous question or was it a stand-alone question?
Figure 1 on Page 3: It is an insightful figure to know aspects that participants consider necessary prerequisites. Was there more information collected about the other option for this question? I am curious to see if some insights can be gained through free text input if it is available. If no free text was collected, please disregard the comment.
Lines 112-113: It might be beneficial to rephrase the first two sentences to provide clear differentiation between the introduction to the section and the question that was in the survey. It is currently unclear whether both the sentences were part of the survey or only the second one.
Figure 2, Page 4: For the labels used in the figure, it will be valuable for readers to know what distinct options entail. Some of the options are self-explanatory, however, some of them can use further elaboration. Can you please provide further clarification for terms used like handling, infrastructure, and patient monitoring used in the labels for the figure? Additionally, when the information was collected on these questions, it would be useful to know if the medical students were provided with more information about these labels in terms of what they mean or if it was based on their understanding of the phrase. This information can further aid readers in interpretation of the results.
Figures 2 and 3. Descriptions for each figure should be further elaborated to differentiate between the concepts being evaluated. To further clarify, figure 2 refers to the problems of digitization in medicine and figure 3 refers to the extent each of the points represents a problem in relation to the digitalization of medicine. Are the ‘problems of digitalization in medicine’ same as or different from the ‘problem in relation to digitalization in medicine’? This clarification can further aid readers in the interpretation of the results.
Lines 148-149, 153-155. It is especially valuable information about the ambiguity faced by the medical students when utilizing computer diagnosis. It clearly brings out an important gap and a need for ensuring that the necessary laws and guidelines are developed for such scenarios.
Lines 155-156: The statement can create confusion as to whether it is a result or a question.
Discussion section:
The discussion section discusses many of the important results presented in the results section. These are extremely important considerations that authors are bringing forth and will have very impactful input for the solution development towards many of the unmet needs that have been identified. The authors provide further details into the topics of result section through referring to pertinent findings and the efforts currently being undertaken to gain insights and develop solutions. Digital transformation is inevitable but being proactive about it and ensuring that the future medical professionals are ready is a key step in the right direction. Considering opinions, suggestions, and concerns of medical professionals for development of solutions is an important and necessary part of the process. I would re-iterate that even though the current study highlights the findings in the Germany faculties and universities, it will equally to pertinent to other regions in general.
The survey results were carried out before the pandemic when digital transformation was underway but the pace of it changed tremendously during the pandemic as many of the facilities were forced to adapt to the changes, especially for the in-person interactions. As a result, many medical professionals had to adapt to the changing requirements and learn new processes in a short duration of time. However, it is unclear how many of the measures implemented during the pandemic lockdowns are used at the same level of operations and if patients and medical professionals prefer in-person interaction. Adding some of those considerations in the discussion sections to make the readers aware of them can help them interpret the results better. In future similar surveys can shed light on how much the pandemic had an impact on the core areas discussed in this manuscript.
Overall, it is a highly informative manuscript with actionable insights for the development of solutions that generate compatible interface between various fields.
Reviewer 4 Report
General: Thank you for the opportunity to review this manuscript. This study analyzed the degree of preparation of digital medicine for medical students in Germany from an online survey. It found that the majority of students stated that they had not taught courses of digitalization. I like this manuscript and believe that it provides the needs of training in digital topics and concerns of medical students for digitalization in medicine.
Major Issues
-2. Material and methods. “The special part is made up of the topics digitalization in medicine, medical activity, professional status, and ethics and law,”: I would recommend providing more details about the questionnaire. This section would match the results sections, which make the reader easier to understand the results.
-4. Discussion. I would suggest adding limitations to the study.
Minor Issues
-page 2 line 75 “(n=225)”. I would suggest moving it to the results section.
-page 2 line 85 “most medical students reported (51.1 %) that…”. Is the most mean out of the 10.4% who answered yes? I would suggest adding both numbers and percentage.
-figure 1. Does the other include more detailed information?
-page 3 line 113 “For example, diagnostic tools could take over medical activities”. I would suggest rephrasing this sentence. Is this a question from the survey?
-page 3 line 125 “(59.6%)” . It is a little confusing to mean only for “insufficient training” or all of them.
-page 4 line 129 “Ethics and law in digitalization”. It did not cover the information in figure 3.
Round 2
Reviewer 2 Report
I feel that the author/s have done a great job in writing this revision. I hope the authors can address the following comments.
1. Although the authors provide a specific number of items for the questionnaire, they do not specify the exact source of each question. We suggest the authors list the questionnaire items in a table to show the question's source clearly.
2. The authors add many recent findings in the digital teaching and learning field. The review of medical students' acceptance and general feelings towards digital healthcare is lacking. Furthermore, the literature is not organized logically.
3. The Discussion sounds like mainly the summary of the results, but one cannot find broader references to other research. It should be amended.
4. It sounds somewhat problematic that the Authors have not seen approval from the local ethical committee, which is against the standards of handling ethical issues in study design.
